# PRE-TRAIN GRAPH NEURAL NETWORKS FOR BRAIN NETWORK ANALYSIS

## ABSTRACT

Human brains, controlling behaviors and cognition, are at the center of complex neurobiological systems. Recent studies in neuroscience and neuroimaging analysis have reached a consensus that interactions among brain regions of interest (ROIs) are driving factors for neural development and disorders. Graph neural networks (GNNs) as a powerful tool for analyzing graph-structured data are naturally applied to the analysis of brain networks. However, training of deep learning models including GNNs often requires a significant amount of labeled data. Due to the complicated data acquisition process and restrictions on data sharing, brain network datasets are still small compared to other types of graphs (e.g., social networks, molecules, proteins). Moreover, real clinical tasks (e.g., mental disorder analysis) are often conducted on local datasets with even smaller scales and larger noises. To this end, we propose to leverage pre-training to capture the intrinsic brain network structures regardless of specific clinical outcomes, for obtaining GNN models that are easily adaptable to downstream tasks. Specifically, (1) we design brain-network-oriented unsupervised pre-training techniques to utilize large-scale brain imaging studies without highly relevant task labels; (2) we develop a data-driven parcellation atlas mapping pipeline to facilitate effective knowledge transfer across studies with different ROI systems. The proposed framework is validated with various GNN models, with extensive empirical results demonstrating consistent improvement in performance as well as robustness.

## 1 INTRODUCTION

In recent years, the analysis of brain networks has attracted considerable interest in neuroscience studies. Brain networks are essentially graphs, where anatomical regions of interest (ROIs) given a parcellation atlas are formed into nodes, and the connectivities among ROIs are formed into edges. Based on brain networks constructed from different modalities such as Diffusion Tensor Imaging (DTI) and functional Magnetic Resonance Imaging (fMRI), effective graph analysis plays a pivotal role in understanding the biological structures and functions of complex neural systems, which can be helpful in the early diagnosis of neurological disorders and facilitate neuroscience research Martensson et al. (2018); Yahata et al. (2016); Lindquist (2008); Smith (2012).

Deep learning has replenished the fields of computer science and beyond. Among various modern deep learning models, the emerging graph neural networks (GNNs) have demonstrated superior performance and even plausible interpretability on a variety of network datasets, including social networks, recommender systems, knowledge graphs, protein and gene networks, molecules and so forth Kipf & Welling (2017); Hamilton et al. (2017); Schlichtkrull et al. (2018); Vashishth et al. (2020); Xu et al. (2019); Ying et al. (2018); He et al. (2020b); Zhang et al. (2020); Liu et al. (2022); Xiong et al. (2020), due to its powerful representations and efficient computations of complex graph structures towards specific downstream tasks. Such achievements on other types of networked data propel studies on GNNs for brain networks, especially models for graph-level classification/regression Ying et al. (2018); Xu et al. (2019); Errica et al. (2020) and important vertex/edge identification Ying et al. (2019); Luo et al. (2020); Vu & Thai (2020), towards tasks such as connectome-based disease prediction and multi-level neural pattern discovery. However, the training of powerful deep learning models including GNNs often requires significant amounts of labeled data Hu et al. (2020a); You et al. (2020); Zhu et al. (2021a). For brain network analysis, there are limited big imaging datasets from a few large-scale national neuroimaging studies such as the ABCD Casey et al. (2018), ADNI

Hinrichs et al. (2009), and PPMI Aleksovski et al. (2018), but such datasets are still rather small compared with graph datasets in other domains (e.g., datasets with 41K to 452K graphs on OGB Hu et al. (2020b) and datasets with thousands to millions of graphs on NetRepo Rossi & Ahmed (2016)).

One solution toward data scarcity is transfer learning which transfers models trained on the large-scale brain network datasets onto small-scale local studies while retaining favorable performance. However, one limitation of transfer learning is its reliance on the availability of similar tasks as supervision during training in the source dataset. In reality, similar tasks in the smaller local studies may not always be available in the large-scale public studies. Pre-training has shown its effectiveness in the fields of computer vision He et al. (2020a); Chen et al. (2020b), natural language processing Devlin et al. (2019b); Radford et al. (2018), as well as graph mining Sun et al. (2022). We explore GNN pre-training on brain networks without supervision and study its effectiveness in the prediction of specific clinical outcomes. However, unique challenges impede the direct application of existing GNN pre-training paradigms to brain networks. For example, brain networks within one study usually share the same node system, which is not properly exploited, whereas different studies often use different node systems, which hinders the transferability of pre-trained models.

To fully unleash the power of GNNs for brain network data by collaborating across datasets, we propose to develop an unsupervised multi-dataset GNN pre-training framework for brain networks. Specifically, we adapt the popular data-efficient framework of MAML with carefully designed two-level contrastive learning that works in concert with the functional neural system modules in brain networks. In addition, to address the problem of dataset misalignment, we propose a novel data-driven atlas mapping technique based on an auto-encoder that transforms the original features in multiple datasets into low-dimensional representations in a uniform embedding space. The transformed features are aligned via variance-based projection with locality-preserving, module-aware, and sparsity-oriented regularizations.

In summary, our contributions are three-folded:

- We identify the intrinsic data-scarcity issue in brain network learning and formulate our problem into an unsupervised pre-training objective.
- We propose a novel brain-network-oriented two-level contrastive sampling strategy for multi-dataset GNN pre-training (Section 3.2). In addition, we implement a data-driven brain atlas mapping strategy with customized regularizations and variance-based sorting to facilitate cross-dataset model sharing (Section 3.3).
- Extensive experiments are conducted to compare our proposed framework with various baselines (Section 4.1), and we further the analysis through in-depth studies on the contributions of each constituent component of our framework (Sections 4.2, 4.3, 4.4).

## 2 RELATED WORK

**Unsupervised Graph Representation Learning.** Most graph representation learning framework often leverages specific label knowledge as ground truth references in the objective function. Unfortunately, brain connectome data often lacks sufficient expert annotation making most supervised optimizations underperform. Therefore, we adopt unsupervised learning techniques for GNN training on brain networks. Motivated by context-based token embedding techniques such as Skip-Gram and Continuous Bag-of-Words (CBOW) Mikolov et al. (2013), Node2vec Grover & Leskovec (2016) and DeepWalk Perozzi et al. (2014) propose to learn node representation by leveraging similarity w.r.t its random-walk induced neighbors. However, this framework is transductive and thus cannot be properly pre-trained and adapted to a new dataset. VGAE Kipf & Welling (2016) proposes to learn representations capable of reconstructing the graph connectivity. However, this design may still adapt poorly to tasks other than link prediction. Recently, the advancement in visual representation learning Chen et al. (2020a); He et al. (2020a) has inspired contrastive learning on graphs. For instance, GBT Bielak et al. (2022) designs a Barlow Twins Zbontar et al. (2021) loss function based on empirical cross-correlation of node embeddings learned from a pair of augmented graph views Zhao et al. (2021). Similarly, GraphCL You et al. (2020) contrasts among graph representations learned from different graphs. Generally, graph contrastive learning captures robust latent representations on well-attributed graphs with a sufficient number of graph samples. However, its accountability may be undermined in sample-scarce and attribute-lacking brain networks.

**Graph Neural Networks Pre-training.** Pre-training of deep neural networks has proven immense success in computer vision Cao et al. (2020); Grill et al. (2020), natural language processing Devlin et al. (2019a); Radford et al. (2018), and multi-modality (e.g. text-image pair) learning Radford et al. (2021); Li et al. (2022); Yao et al. (2022); Wang et al. (2022). Specifically, the pre-training pipeline is formulated by defining pretext tasks such as masked token prediction in NLP or image recovery from Gaussian blurring in CV. Following a similar intuition, GPT-GNN Hu et al. (2020c) proposes graph-oriented pretext tasks, for instance, masked attribute and edge prediction. L2P-GNN Lu et al. (2021) introduces dual-adaptation by simultaneously optimizing the encoder on a node-level link prediction objective and a graph-level self-supervision task similar to DGI Velickovic et al. (2019). Interestingly, PHD Li et al. (2021a) leverages half-graph representations and optimizes on a connected component prediction objective. Others, such as GMPT Hou et al. (2022) devises inter-graph message passing to obtain contextual node embedding and optimizes concurrently under supervision and self-supervision. Overall, most GNN pre-training strategies emphasize the effective formulation of pretext tasks and data augmentation. However, existing techniques cannot be directly localized to brain networks because they do not consider the special properties of brain connectome data.

## 3 UNSUPERVISED BRAIN NETWORK PRE-TRAINING

**Problem Definition.** Formally, given a collection of brain network datasets $\mathcal{S} = \{\mathcal{D}_1, \mathcal{D}_2, \cdots \mathcal{D}_s\}$, each $\mathcal{D}_i = \{\mathcal{G}_{i,j}\}_{j=1}^{N_i}$ is composed of $N_i$ subjects of weighted graphs $\mathcal{G}_{i,j}$. Specifically, each graph $\mathcal{G}_{i,j}$ is described by a node set $\mathcal{V}_{i,j} = \{v_m\}_{m=1}^{M_{i,j}}$, an edge set $\mathcal{E}_{i,j} = \mathcal{V}_{i,j} \times \mathcal{V}_{i,j}$, and a weighted adjacency $\boldsymbol{A}_{i,j} \in \mathbb{R}^{M_{i,j} \times M_{i,j}}$. We define a $\theta$ parameterized GNN $f(\cdot)$ that learns a mapping relation $f_\theta(\mathcal{G}_{i,j}) = z_{i,j}$ where $z_{i,j}$ denotes the learned representation(s). The goal of our framework focuses on pre-training $f(\cdot)$ on a subset $\mathcal{S}_{\text{source}}$ of $\mathcal{S}$ via self-supervision, to obtain an initialization $\theta_0$ that can efficiently converge toward a target optimal $\theta^*$ on a dataset $\mathcal{D}_{\text{target}} \in \mathcal{S}$.

### 3.1 MULTI-DATASET GNN PRE-TRAINING

Model pre-training aims at finding proper initialization that can be efficiently adapted toward downstream evaluations. Specifically, in our framework, the model is first jointly pre-trained on multiple brain network datasets under self-supervision, and later applied to a target dataset with a supervised objective. Importantly, our pipeline differs from traditional transfer learning since it requires measurable similarity between the source and target data as well as their learning objectives, whereas transferable task formulations are often missing for brain network analysis.

To enable faster convergence, our multi-dataset pre-training adapts the popular data-efficient training techniques presented in MAML Finn et al. (2017). Specifically, at each training iteration, we partition each input dataset into an inner-loop support set and an outer-loop query set. The model is first trained on the support set without explicitly updating the parameters. Instead, we temporarily store the updates as fast weights Ba et al. (2016), which we later evaluate on the query set to compute the actual gradients. In other words, the training makes use of approximating higher-order derivatives Tan & Lim (2019) at each step to achieve a quicker descent along the optimization trajectory. In our scenario, the joint optimization sums over the loss on each brain network dataset, that is, for $n$ number of datasets along with their respective temporary fast weights $\theta'$ and query partition, the step-wise update of the model parameter at time $t$ states as $\theta^{t+1} = \theta^t - \alpha \nabla_\theta \sum_{i=1}^{n} \mathcal{L}_{\text{query}_i} f_{\theta_i'^t}(\cdot)$.

### 3.2 BRAIN NETWORK ORIENTED TWO-LEVEL CONTRASTIVE LEARNING

Considering the high cost of acquiring informative task labels for brain networks, our pre-training pipeline resorts to a label-free learning strategy, in particular, contrastive learning (CL). Generically, given an anchor point of investigation $X$ from a data distribution $\mathcal{H}$, CL seeks to maximize the mutual information (MI) $I(\cdot; \cdot)$ against its positive samples $X^+$ and minimize MI against its negative samples $X^-$. Formally, we express the cost function as below:

$$\mathcal{J}_{\text{con}} = \mathbb{E} \left[ \frac{1}{|\mathcal{H}|} \sum_{X, X^-, X^- \sim \mathcal{H}} \left( -I(X; X^+) + I(X; X^-) \right) \right]. \tag{1}$$

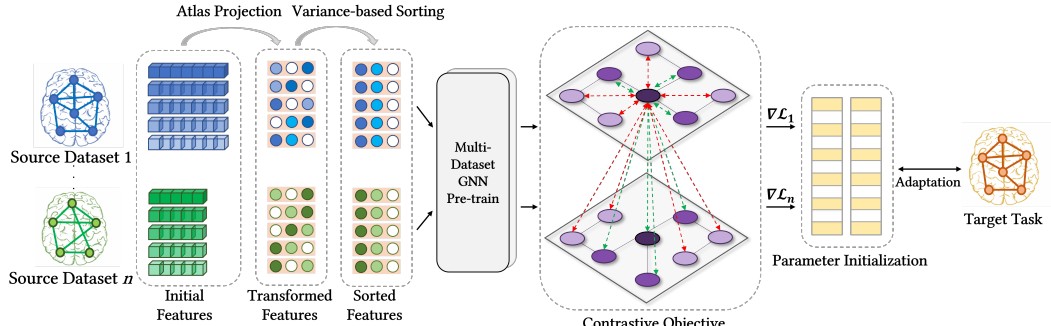

**Figure 1:** Overview of our proposed framework. We first project the source data features into a fixed dimensional representation through atlas transformation followed by variance-based feature alignment. We then pre-train a GNN encoder on multiple datasets under self-supervision via novel two-level contrastive learning. Finally, learned parameters are fine-tuned on a target task.

Applying this concept toward graph-level CL, given an anchor node representation $z_\alpha$ and an arbitrary set $\mathbf{S}$ containing a collection of node representations sampled from the training batch. Based on the Jensen-Shannon divergence, the MI between $z_\alpha$ and the representations in $\mathbf{S}$ is elaborated as:

$$I\left(z_\alpha; \mathbf{S}\right) = \sum_{z_s \in \mathbf{S}} \mathrm{JSD}(z_\alpha; z_s) = \frac{1}{2|\mathbf{S}|} \sum_{z_s \in \mathbf{S}} \log\left(1 + \exp\left(\frac{z_\alpha^T z_s}{\|z_\alpha^T z_s\|} + \frac{z_s^T z_\alpha}{\|z_s^T z_\alpha\|}\right)\right), \quad (2)$$

The end goal of our framework focuses on localizing effective GNN CL learning Zhu et al. (2020; 2021b) for brain networks. To this end, suppose a dataset $\mathcal{D}$ and an anchor node $i$ from graph $\mathcal{G}_p \in \mathcal{D}$ with the learned representation $z_{i,p}$, we propose to categorize the possible sample selections into three fundamental types (a gentle visualization is shown in Figure 2):

- $\underline{\mathbf{S_1}}$: $\{z_{j,p} : j \in \mathcal{N}_k(i,p)\}$, which refers to the set of node representations within the anchor's $k$-hop neighborhood in graph $\mathcal{G}_p$.
- $\underline{\mathbf{S_2}}$: $\{z_{j,p} : j \notin \mathcal{N}_k(i,p)\}$, which refers to the set of remaining node representations in graph $\mathcal{G}_p$ that are not in the anchor's $k$-hop neighborhood.
- $\underline{\mathbf{S_3}}$: $\{z_{j,q} : \mathcal{G}_q \in \mathcal{D}, j \in \mathcal{G}_q, q \neq p\}$, which refers to the set of node representations in all other graphs in the dataset $\mathcal{D}$.

Notice that our framework leverages the $k$-hop substructure of the anchor node to further differentiate $\mathbf{S_1}$ and $\mathbf{S_2}$ for contrastive optimization. This design is motivated by two perspectives: **(1) Regarding GNN learning.** Since the node representation is learned from information aggregation of its $k$-hop neighborhood, maximizing the MI of an anchor to its $k$-hop neighbors naturally encourages lossless message passing of GNN convolutions. **(2) Regarding the uniqueness of brain networks.** Brain networks can be anatomically segmented into smaller-scaled neural system modules Cui et al. (2022b), thus capturing subgraph-level knowledge can provide useful signals for brain-related analysis.

On top of these three fundamental types of samples, we further introduce an additional type by leveraging the property of brain networks that ROI identities and orders are fixed across samples. Taking this advantage, we can further promote GNN to extract shared substructure knowledge by evaluating the MI of an anchor against its presence in other graphs. Given an anchor representation $z_{i,p}$ of node $i$ from graph $\mathcal{G}_p \in \mathcal{D}$, the novel inter-graph sample type is defined as:

- $\underline{\mathbf{S_4}}$: $\{z_{j,q} : j \in \mathcal{N}_k(i,q) \cap \mathcal{N}_k(i,p), \mathcal{G}_q \in \mathcal{D}, q \neq p\}$, which refers to the set of node representations within the $k$-hop neighborhood of node $i$ in all other graphs in $\mathcal{D}$. Conceptually, $\mathbf{S_4}$ is a special subset of $\mathbf{S_3}$.

Notice that for an anchor node $i$, its $k$-hop neighborhood structures might not be identical among different graphs. Thus, we only consider shared neighborhoods for multi-graph MI evaluation.

To encourage the learning of unique neighborhood knowledge in a single brain network instance, and shared substructure knowledge across the entire dataset, we configure $\mathbf{S_1}$ and $\mathbf{S_4}$ as positive samples while $\mathbf{S_2}$ and the set $\mathbf{S_3} - \mathbf{S_4}$ as negative samples, as illustrated in Figure 3.

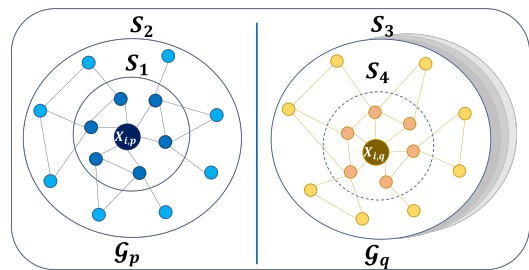

**Figure 2:** Visual example of the sample types where $X_{i,p}$ is the anchor and $\mathbf{S_1}/\mathbf{S_4}$ are sampled as 1-hop neighbors.

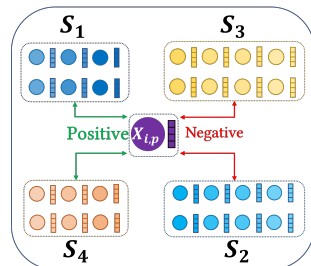

**Figure 3:** Sampling configurations of our framework.

In addition, our sampling categorization can also help understand the objective formulations in various state-of-the-art graph CL frameworks Velickovic et al. (2019); Qiu et al. (2020); Xia et al. (2022); Sun et al. (2019); Zhu et al. (2021a). We hereby summarize our findings in Table 1. Specifically, "+" denotes positive sampling; "-" denotes negative sampling; and "/" means that the sample type is not considered. Observe that since DGI and InfoGraph leverage graph representation pooled from node representations as a special sample, it is essentially equivalent to jointly considering $\mathbf{S_1}$ and $\mathbf{S_2}$ without explicit differentiation. On the other hand, GCC and EGI, which are more intuitively connected to our framework, lever-

**Table 1:** Sampling configurations of existing graph CL methods.

|        | $\mathbf{S_1}$ | $\mathbf{S_2}$ | $\mathbf{S_3}$ | $\mathbf{S_4}$ |
|--------|------|------|------|------|
| DGI    | +    | +    | /    | /    |
| InfoG  | +    | +    | −    | /    |
| GCC    | +    | −    | −    | /    |
| EGI    | +    | −    | −    | /    |
| Ours   | +    | −    | −    | +    |

age neighborhood MI maximization on a single graph, but fail to extend this consideration to a multi-graph setting like ours.

## 3.3 Data-driven Brain Atlas Mapping

**Motivation.** When a pre-trained model is fine-tuned on a new data domain, misalignment between source and target signals may negatively affect downstream adaptation. This problem is especially concerning in brain networks where datasets are built from various brain atlas templates each mapping to a unique system of ROIs. For example, the HIV dataset is parcellated from the AAL90 template, resulting in 90 defined ROIs; whereas the PPMI dataset adopts the Desikan-Killiany84 template, resulting in 84 defined ROIs. Thus, brain networks in the two datasets will have different numbers and semantics of nodes. Although GNNs can naturally handle graphs without fixed numbers and orders of nodes, when we construct the most informative ROI features as the brain connection profiles Cui et al. (2022a), the feature dimensions and their meanings will be different. Although manual conversion can be done to translate among various templates, an accurate one-to-one mapping is impossible, and even obtaining a coarse mapping still needs tremendous expert effort. To this end, to facilitate an effective target adaptation, we propose a data-driven atlas mapping solution that transforms the original node features into lower-dimensional embedding spaces that not only preserves original connectivity information but also achieves cross-dataset feature alignment.

### 3.3.1 Autoencoder with Brain Network Oriented Regularizers

To address information preservation, we adopt a one-layer linear autoencoder (AE) as our base structure. Specifically, AE features a linear projection encoder $\mathbf{W}$ and a transposed decoder $\mathbf{W}^T$. The general objective is to learn a low-dimensional projection that can easily reconstruct to its original presentation. Formally, the loss function is defined as minimizing the reconstruction $\mathcal{L}_{\text{rec}} = (1/n)\|\mathbf{X} - \mathbf{X}\mathbf{W}\mathbf{W}^T\|_2^2$ for an input $\mathbf{X} \in \mathbb{R}^{n \times d}$ and a projection $\mathbf{W} \in \mathbb{R}^{d \times m}$ Hinton & Zemel (1993). To achieve a more informative feature compression besides dimension reduction and to guide the overall AE optimization, we propose to implement several regularizers that take special consideration of brain network characteristics:

**Locality-Preserving Regularizer.** Following section 3.2, a significant portion of brain network analysis focuses on learning neighborhood structure. Naturally, the compressed feature should preserve the neighborhood information from its original profile. Considering that each ROI atlas contains a unique set of 3D coordinates specifying the scaled actual locations of ROIs in the human

brain, one important type of neighborhood is its physical adjacencies based on the 3D projection. Hence, we regularize our AE objective as neighborhood preserving He et al. (2005), which can be written as $\mathcal{L}_{\text{loc}} = (1/n)\|\mathbf{Y} - \mathbf{TY}\|^2$, where $\mathbf{Y} \in \mathbb{R}^{n \times m}$ denotes the AE projected features and $\mathbf{T} \in \mathbb{R}^{n \times n}$ is the transition matrix built from the $k$-NN graph of the 3D coordinates.

**Module-Aware Regularizer.** Brain networks can be segmented into various neural system modules each characterizing a subset of ROIs. In graph terminology, they are community structures. Given this, the projected feature could also incorporate information regarding neural system belongings. Unfortunately, knowing such ground-truth segmentations requires significant prior knowledge. Hence, we resort to community detection methods on graphs, and in particular, our design follows the modularity maximization paradigm in which the regularizer Salha-Galvan et al. (2022) can be formally written as minimizing

$$\mathcal{L}_{\text{com}} = -\frac{1}{2m} \sum_{i,j=1}^{n} \left[ \mathbf{A}_{ij} - \frac{k_i k_j}{2m} \right] \exp(-\|y_i - y_j\|_2^2), \tag{3}$$

where $\mathbf{A} \in \mathbb{R}^{n \times n}$ denotes graph adjacency, $k_i$ denotes degree of node $i$, and $y_i$ denotes the projected features. Essentially, the optimization minimizes the $L_2$ distance of representations among nodes within the same communities measured by the modularity and maximizes the distance otherwise.

**Sparsity-Oriented Regularizer.** Sparse networks have demonstrated robustness in learning representations from noise Jeong et al. (2017); Shi et al. (2019); Makhzani & Frey (2014) and it has been shown that, in brain connectome analysis, sparsity leads to a better interpretation of task-specific ROI connections in generation and classification tasks Kan et al. (2022). To this end, we adopt the popular KL-divergence smoothing to enforce parameter sparsity, which is given as:

$$\mathcal{L}_{\text{KL}} = \sum_{i=1}^{n} \sum_{j=1}^{d} \left[ \rho \log \left( \frac{\rho}{\hat{\rho}_{ij}} \right) + (1 - \rho) \log \left( \frac{1 - \rho}{1 - \hat{\rho}_{ij}} \right) \right], \tag{4}$$

where $\rho$ is the target sparsity value usually set to a small positive float, and $\hat{\rho}_{ij}$ is the element-wise activation of the $n \times d$ encoder projection matrix.

### 3.3.2 Variance-based Dimension Sorting

In addition to dataset-specific feature transformation, cross-dataset alignment of feature signals also greatly facilitates model adaptation. Taking a closer look at the one-layer AE transformed feature vectors, essentially, each dimension in the projected space is formed by a weighted (nonlinear) combination of multiple dimensions in the original space. Localizing this concept to brain networks, the process is ultimately grouping the ROIs and their signals to derive new feature dimensions, which can be regarded as *virtual ROIs*. Although different datasets devise unique atlas templates, the base brain anatomy construction process follows fundamental theorems and practices in the neuroscience literature Brodmann (1909); Zhou et al. (2020). The final variations in ROI parcellations are results of post-processing due to different analytical and clinical conventions. Therefore, it is reasonable to assume a shared virtual ROI system underlying different parcellation systems. Incorporating the objectives of community learning and neighborhood preserving regularizers introduced in section 3.3, we are essentially training the AE to capture such shared virtual ROIs from a data-driven perspective. Our final step is to align the discovered virtual ROIs across datasets such that, regardless of origins, each virtual ROI characterizes the same functional unit in the human brain.

Observe that a one-layer linear AE shares a similar objective with PCA (more details in Appendix A.1)but with greater customizability (*i.e.,* adding regularizers). In particular, PCA automatically orders the dimensions based on decreasing levels of sample variance Hotelling (1933). Taking this inspiration, we can leverage the learned parameters of the AE projection to approximate a variance estimate for each virtual ROI (*i.e.,* projected feature dimension). Intuitively, each virtual ROI is computed with the sample variance indicating its representativeness of the original data variations. Based on the previous discussion on shared patterns across different parcellation systems, we assume similar virtual ROIs in different datasets with different atlas templates would give similar variance scores, at least regarding the order of the scores. Thus, by separately sorting the same number of virtual ROIs according to the sample variance in each dataset, we expect to achieve a cross-dataset virtual ROI alignment, regarding the actual functional unit in the human brain. We summarize this procedure in detail in Appendix A.2 Algorithm 1.

## 4 EXPERIMENTS

We analyze our framework by conducting extensive experiments on real brain network datasets, with a focus on these research questions: **RQ1**: How does our method perform compared to other popular unsupervised GNN pre-training frameworks adapted to our setting? **RQ2**: How does each major component in our framework contribute to the overall performance? **RQ3**: How do the difference in sampling consideration influence model convergence and adaptation performance? **RQ4**: How does our variance-based sorting align virtual ROIs among different parcellation systems?

**Datasets, configurations, and metrics.** Our experiments use brain network datasets of diverse modalities from three IRB approved sources, including Parkinson's Progression Markers Initiative (PPMI), Bipolar Disorders (BP), and Human Immunodeficiency Virus Infection (HIV). Specifically, the PPMI is a public dataset that includes brain networks constructed from three tractography algorithms namely the Probabilistic Index of Connectivity (PICo), Hough voting (Hough), and FSL for 718 subjects, where 569 subjects are Parkinson's Disease (PD) patients and the rest 149 are Healthy Control (HC) ones; while BP is a private dataset composed of the resting-state fMRI and DTI image data of 52 Bipolar I subjects who are in euthymia and 45 HCs, and HIV involves 70 subjects, with 35 early HIV patients and 35 HCs. More details about the dataset origin and preprocessing are provided in Appendix B. We pre-train the model on the PPMI data where brain networks constructed from each tractography algorithm are treated as a separate pre-training dataset and then evaluate the target performance on BP and HIV. For backbone selection, we employ GCN as our base GNN encoder. Detailed hyper-parameter settings regarding the entire pipeline and GCN setup are described in Appendix C. In addition, we also benchmark our framework on GAT and GIN, and the performance is reported in Appendix D.1. Our target evaluation of disease prediction is formulated as a binary graph classification problem. To this end, we adopt the two widely used metrics in the medical discipline Li et al. (2021b); Cui et al. (2022a), namely, accuracy score (ACC) and the area under the receiver operating characteristic curve (AUC) to assess the downstream performance.

### 4.1 OVERALL PERFORMANCE COMPARED WITH BASELINES (RQ1)

We present a comprehensive comparison of the target performance between our framework and existing unsupervised baselines in Table 2. The baselines are grouped based on their optimization strategies: **Group 0**: No pre-training (NPT), where model parameters are randomly initialized before fine-tuning on the target. **Group 1**: Non-CL-based frameworks, including Node2Vec, DeepWalk, and VGAE, whose cost functions focus on co-occurrence agreement or link reconstruction. **Group 2**: Single-scale CL, including GBT, ProGCL, and GraphCL, whose CL optimizations leverage either node- or graph-level representations. **Group 3**: Multi-scale CL, including DGI and InfoGraph (InfoG), whose CL optimizations depend on examining both nodes- and graph-level representations. **Group 4**: Ego-graph sampling, including GCC and EGI, whose contrastive samplings consider $k$-hop ego-networks as discriminative instances, which are the most similar to our method.

**Table 2:** Disease prediction performance comparison. All results are averaged from 5-fold cross-validation along with standard deviations and have passed the significance test with $p = 0.05$.

| Method | BP-fMRI | | BP-DTI | | HIV-fMRI | | HIV-DTI | |
|---|---|---|---|---|---|---|---|---|
| | ACC | AUC | ACC | AUC | ACC | AUC | ACC | AUC |
| NPT | $50.07_{\pm13.70}$ | $50.11_{\pm15.48}$ | $49.51_{\pm14.68}$ | $51.83_{\pm13.98}$ | $56.27_{\pm15.84}$ | $57.16_{\pm15.14}$ | $51.30_{\pm16.42}$ | $53.82_{\pm14.94}$ |
| Node2Vec | $48.51_{\pm10.39}$ | $49.68_{\pm7.23}$ | $50.83_{\pm8.14}$ | $46.70_{\pm10.33}$ | $52.61_{\pm10.38}$ | $50.75_{\pm10.94}$ | $49.65_{\pm10.30}$ | $51.22_{\pm10.79}$ |
| DeepWalk | $50.28_{\pm9.33}$ | $51.59_{\pm9.06}$ | $52.17_{\pm9.74}$ | $48.36_{\pm9.37}$ | $54.81_{\pm11.26}$ | $55.55_{\pm11.93}$ | $52.67_{\pm11.42}$ | $50.88_{\pm10.53}$ |
| VGAE | $56.71_{\pm9.68}$ | $55.24_{\pm11.48}$ | $54.63_{\pm12.09}$ | $54.21_{\pm11.94}$ | $62.76_{\pm9.47}$ | $61.25_{\pm11.61}$ | $56.90_{\pm9.72}$ | $55.35_{\pm9.04}$ |
| GBT | $57.21_{\pm10.68}$ | $57.32_{\pm10.48}$ | $56.29_{\pm9.35}$ | $55.27_{\pm10.54}$ | $65.73_{\pm10.93}$ | $66.08_{\pm10.43}$ | $59.80_{\pm9.76}$ | $57.37_{\pm9.49}$ |
| GraphCL | $59.79_{\pm9.36}$ | $59.10_{\pm10.78}$ | $57.57_{\pm10.63}$ | $57.35_{\pm9.67}$ | $67.08_{\pm9.70}$ | $69.17_{\pm10.68}$ | $60.43_{\pm8.39}$ | $60.03_{\pm10.48}$ |
| ProGCL | $61.36_{\pm8.90}$ | $61.61_{\pm9.34}$ | $60.26_{\pm8.37}$ | $61.67_{\pm8.46}$ | $70.52_{\pm9.19}$ | $71.16_{\pm9.85}$ | $61.48_{\pm10.38}$ | $60.94_{\pm10.57}$ |
| DGI | $62.44_{\pm10.12}$ | $60.75_{\pm10.97}$ | $58.15_{\pm9.63}$ | $58.95_{\pm9.60}$ | $70.22_{\pm11.43}$ | $70.12_{\pm12.46}$ | $60.83_{\pm10.84}$ | $62.06_{\pm10.16}$ |
| InfoG | $62.87_{\pm9.52}$ | $62.37_{\pm9.67}$ | $60.88_{\pm9.97}$ | $60.44_{\pm9.61}$ | $72.46_{\pm8.71}$ | $72.94_{\pm8.68}$ | $61.75_{\pm9.76}$ | $61.37_{\pm9.85}$ |
| GCC | $63.45_{\pm9.82}$ | $62.39_{\pm9.08}$ | $60.44_{\pm9.54}$ | $60.29_{\pm10.33}$ | $70.97_{\pm10.31}$ | $72.48_{\pm11.36}$ | $61.27_{\pm9.66}$ | $61.38_{\pm10.72}$ |
| EGI | $63.38_{\pm8.93}$ | $63.58_{\pm8.02}$ | $61.82_{\pm8.53}$ | $61.57_{\pm8.27}$ | $73.46_{\pm8.49}$ | $73.28_{\pm8.68}$ | $60.89_{\pm9.87}$ | $62.41_{\pm8.50}$ |
| Ours | $\mathbf{68.84_{\pm8.26}}$ | $\mathbf{68.45_{\pm8.96}}$ | $\mathbf{66.57_{\pm7.67}}$ | $\mathbf{68.31_{\pm9.39}}$ | $\mathbf{77.80_{\pm9.76}}$ | $\mathbf{77.62_{\pm8.74}}$ | $\mathbf{67.51_{\pm8.67}}$ | $\mathbf{67.74_{\pm8.59}}$ |

Overall, we highlight the following observations:

- Our framework demonstrates consistent improvements over the baselines. In particular, our method achieves 7.34%-13.30% relative gain over the best performing baselines and 31.80%-38.26% relative gain over the NPT setting across all metrics.
- Compared with the transductive methods of Node2Vec and DeepWalk, the GNN pre-trained by VGAE learns structure-preserving representations and achieves the most promising results in group 1, but the performance is still visibly inferior to Ours.
- Maximizing MI between augmentations from a singular instance may prevent GNN from learning shared knowledge about the entire dataset. For baselines in groups 2-4, non-augmentation-based CL pre-training generally reports 4.36% relative gain across both metrics and a 7.63% relative decrease in performance variance compared to augmentation-based counterparts. For this reason, Ours does not deploy data augmentation.
- Multi-scale MI encourages capturing of effective local (*i.e.*, node-level) representations capable of summarizing global (*i.e.*, graph-level) information of the entire network. Hence, on average, group 3 outperforms group 2 by a relative gain of 2.68% in ACC and 3.27% in AUC.
- Group 4 presents the strongest baseline which benefits from its considerations of $k$-hop neighborhood, indicating the importance of local neighborhoods in brain network analysis. Ours is the only method that captures nodes' local neighborhoods through both nodes- and graph-level CL.

## 4.2 ABLATION STUDIES (RQ2)

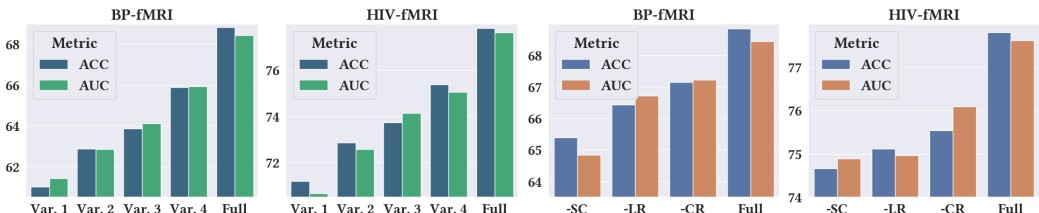

**Figure 4:** Ablation comparisons on contrastive sampling considerations (left two) and atlas mapping regularizers (right two). The $y$-axis refers to the numeric values of evaluated metrics (in %).

Figure 4 shows the contribution of each constituent part of our framework to the overall performance in the fMRI view (the DTI version is in Appendix D.2). Specifically, the two sub-figures on the left investigate the different sampling considerations of $S_1 - S_4$. In particular, we list four possible variants detailed in Table 3. To this end, we deliver three insights regarding the comparison: **(1)** leveraging $k$-hop neighborhood (*i.e.,* positive $S_2$) MI maximization brings visible performance gain, confirming its benefit in brain structure

**Table 3:** Sampling variants overview

|  | $S_1$ | $S_2$ | $S_3$ | $S_4$ |
|---|---|---|---|---|
| Var. 1 | – | – | / | / |
| Var. 2 | + | – | / | / |
| Var. 3 | + | – | – | / |
| Var. 4 | + | + | – | / |

learning; **(2)** extending to multi-graph CL (*i.e.,* consideration of $S_3$) encourages extraction of unique ROI knowledge, thus leading to improved results in Var. 3/4; **(3)** Var. 4 performs better than Var. 3 likely due to its summarization of global (*i.e.,* graph-level) information in local node representations.

Meanwhile, the two sub-figures on the right focus on atlas mapping regularizers by comparing the downstream results without the sparsity regularizer (-SR), the locality regularizer (-LR), and the community regularizer (-CR) respectively against our full framework. We hereby remark two key observations: **(1)** removing SR leads to the greatest performance drop, indicating its importance in learning noise robust projections; **(2)** LR and CR are both crucial to atlas mapping optimization, validating our intuition in learning neighborhood-aware and blockwise feature information.

## 4.3 IN-DEPTH ANALYSIS OF TWO-LEVEL CONTRASTIVE SAMPLING (RQ3)

Figure 5 shows the further investigation of the four sampling variants and the full framework in terms of pre-training convergence, target adaptation progression, and pre-training runtime consumption. Overall, we highlight three important observations: **(1)** From Figure 5(a), we can observe that the pre-training convergence is generally efficient across all variants with the MAML-inspired multi-dataset joint optimization technique. In addition, the full model presents the most optimal convergence confirming the benefit of learning shared neighborhood information in brain network data

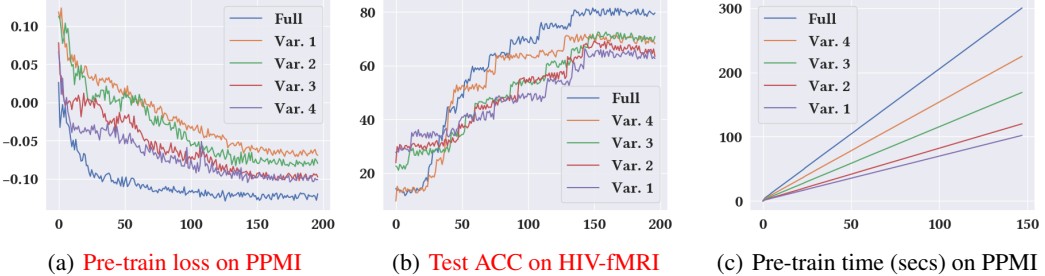

(a) Pre-train loss on PPMI     (b) Test ACC on HIV-fMRI     (c) Pre-train time (secs) on PPMI

**Figure 5:** In-depth comparison among the four variants and our full model. The $x$-axis is epochs.

via two-level node contrastive sampling. **(2)** From Figure 5(b), the effectiveness of learning shared substructure knowledge within a fixed ROI system is directly reflected in a superior downstream adaptation compared to other variants. **(3)** From Figure 5(c), the more sophisticated the sampling considerations, the greater the computational complexity for MI evaluation, resulting in larger runtime for each pre-training epoch. However, the total time consumptions are all on the same scale.

## 4.4 IN-DEPTH ANALYSIS OF VIRTUAL ROI ALIGNMENT (RQ4)

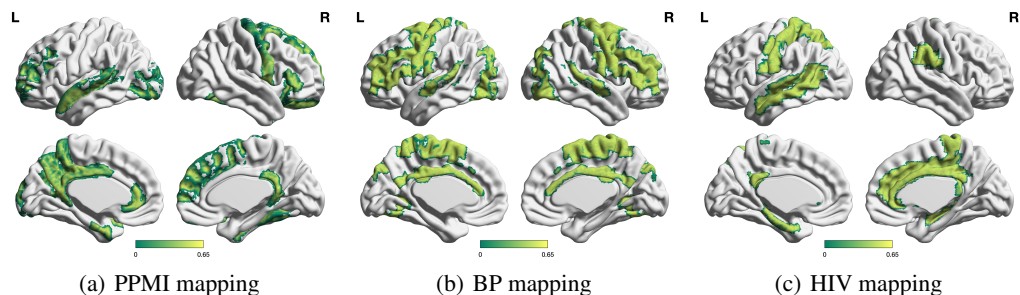

(a) PPMI mapping        (b) BP mapping        (c) HIV mapping

**Figure 6:** Virtual ROI Mapping across the three investigated datasets.

To further validate our designs of variance-based virtual ROI sorting, for each atlas template (*i.e.,* dataset), we select the top 2 virtual ROIs that have the highest sample variances and backtrack to locate their projected ROIs. We then illustrate in Figure 6 a visualization highlighting the original ROIs on a 3D brain surface. Specifically, we remark on two key takeaways: **(1)** The three atlases share similar ROI projection regions in the medial (*i.e.,* bottom left and right) views of the brain surface, reflecting the working effectiveness of our ROI alignment design. **(2)** There exist certain variations among ROI projections from the lateral (*i.e.,* top left and right) view, which shows a limitation of unsupervised atlas mapping but is still better than no alignment at all.

## 5 CONCLUSION

Brain network analysis for task-specific disease prediction is challenging for conventional GNN frameworks. Firstly, effective knowledge extraction for deep neural models requires a significant amount of training data and task labels which are often lacking in neuroimaging studies, especially for local downstream tasks. Moreover, the absence of a unifying brain atlas definition prevents efficient knowledge transfer across different domains. Through extensive benchmarking on real-world brain connectome datasets, our framework reports superior and robust performance in disease prediction and clearly outperforms various state-of-the-art baselines. Specifically, our framework implements unsupervised multi-dataset GNN pre-training via two-level node contrastive sampling to address the data-scarcity concern. Concurrently, our framework also features atlas mapping via brain-network-oriented regularizers and variance-based sorting which presents a data-driven solution to the incompatible ROI parcellation systems for cross-dataset model adaptation. Due to the difficulties in obtaining many brain network datasets, our experiments are focused on three widely used brain imaging studies, and it is natural to further validate the generalizability of our framework with more brain network datasets when they become available.

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

## A    ANALYSIS OF THE AUTOENCODER STRUCTURE

### A.1    BRIDGING RECONSTRUCTION MINIMIZATION AND VARIANCE MAXIMIZATION

In this subsection, we briefly discuss how the reconstruction minimizing objective in one-layer AE can be cast to a variance maximizing objective in PCA. Assume given a data matrix $\mathbf{X} \in \mathbb{R}^{n \times d}$, its covariance matrix $\mathbf{\Sigma} = \mathbf{X}^T \mathbf{X} \in \mathbb{R}^{n \times n}$, and a single-layer AE projection matrix $\mathbf{W} \in \mathbb{R}^{d \times m}$ with parameters randomly initialized from the continuous uniform distribution $\mathcal{U}(0, 1)$, the reconstruction objective is:

$$
\begin{aligned}
\frac{1}{n}\|\mathbf{X} - \mathbf{X}\mathbf{W}\mathbf{W}^T\|^2 &= \frac{1}{n}\mathrm{tr}\left((\mathbf{X} - \mathbf{X}\mathbf{W}\mathbf{W}^T)(\mathbf{X} - \mathbf{X}\mathbf{W}\mathbf{W}^T)^T\right) \\
&= \frac{1}{n}\mathrm{tr}\left((\mathbf{X} - \mathbf{X}\mathbf{W}\mathbf{W}^T)(\mathbf{X}^T - \mathbf{W}\mathbf{W}^T\mathbf{X}^T)\right) \\
&= \frac{1}{n}[\mathrm{tr}(\mathbf{X}\mathbf{X}^T) - \mathrm{tr}(\mathbf{X}\mathbf{W}\mathbf{W}^T\mathbf{X}^T) - \mathrm{tr}(\mathbf{X}\mathbf{W}\mathbf{W}^T\mathbf{X}^T) \\
&\quad + \mathrm{tr}(\mathbf{X}\mathbf{W}\mathbf{W}^T\mathbf{W}\mathbf{W}^T\mathbf{X}^T)] \\
&= \frac{1}{n}[c_1 - 2 \cdot \mathrm{tr}(\mathbf{X}\mathbf{W}\mathbf{W}^T\mathbf{X}^T) + \mathrm{tr}(\hat{\mathbf{X}}\hat{\mathbf{X}}^T)] \\
&= \frac{1}{n}[c_1 - 2 \cdot \mathrm{tr}(\mathbf{X}\mathbf{W}\mathbf{W}^T\mathbf{X}^T) + c_2] \\
&= c_3 - c_4 \cdot \mathrm{tr}(\mathbf{W}^T\mathbf{X}^T\mathbf{X}\mathbf{W}) \\
&= c_3 - c_4 \cdot \mathrm{tr}(\mathbf{W}^T\mathbf{\Sigma}\mathbf{W})
\end{aligned}
$$

Notice that $c_1, c_2, c_3, c_4$ are non-negative scalar constants that do not influence the overall optimization schedule. Hence, alternatively, the optimal AE projection also maximizes the sample variance $\mathrm{tr}(\mathbf{W}^T\mathbf{\Sigma}\mathbf{W})$, achieving an identical end goal of PCA transform. Specifically, according to PCA, variance maximization is realized by constructing the projection $\mathbf{W}$ to contain the set of orthonormal eigenvectors of $\mathbf{\Sigma}$ that gives the largest eigenvalues Hotelling (1933). That is, there is an orthogonality constraint on $\mathbf{W}$. Minimizing the MSE reconstruction also results in an orthogonal $\mathbf{W}$:

$$
\frac{1}{n}\|\mathbf{X} - \mathbf{X}\mathbf{W}\mathbf{W}^T\|^2 = 0 \Rightarrow \mathbf{X}\mathbf{W}\mathbf{W}^T = \mathbf{X} \Rightarrow \mathbf{W}\mathbf{W}^T = \mathbf{I}
$$

Therefore, the optimal AE projection $\mathbf{W}$ is also capturing a set of variance-maximizing orthogonal vectors. Note that the AE optimized $\mathbf{W}$ is theoretically equivalent to the eigendecomposition of $\mathbf{\Sigma}$ if and only if the reconstruction loss is 0. Therefore, in practice, the AE is, at best, an approximate solution to variance maximization.

### A.2    VARIANCE-BASED SORTING PROCEDURE

Following the discussion in A.1, assuming a perfect optimization, the linear one-layer AE behaves similarly to PCA, and there is an equivalence relation between their respective objective functions. Notice that in PCA, the eigenvalue of the covariance matrix $\mathbf{\Sigma}$ signifies the intensity of data variation along the direction of its corresponding eigenvector, which is essentially a column entry of the transformation matrix. Then intuitively, given an optimized AE projection $\mathbf{W}$, we can examine, for each column of $\mathbf{W}$, its representativeness (*i.e.,* data variance) of the data covariance with a scalar estimate (*i.e.,* an eigenvalue-like scoring). Inspired by the properties of eigendecomposition, we can approximate these estimates by measuring the distance of $\mathbf{W}$ w.r.t to the product of linearly transforming $\mathbf{W}$ through $\mathbf{\Sigma}$ by a scaling factor of $\mathbf{\lambda}$. More specifically, we want to solve for $\mathbf{\lambda}$ such that $\mathbf{\Sigma}\mathbf{w} = \mathbf{\lambda}\mathbf{w}$ for every column vector $\mathbf{w} \in \mathbf{W}$. Under the PCA perspective, $\mathbf{\lambda}$ contains the variance estimate for each column-wise individual projection of $\mathbf{W}$. To this end, we detail the sorting procedure in Algorithm 1.

## B    DATASET DETAILS

All datasets used, which are introduced below, are processed anonymously with no personally identifiable information. In addition, all related studies are conducted according to the Good Clinical

---

**Algorithm 1** Overview procedure for variance-based sorting

---

**Input:** Original feature matrix $\mathbf{X} \in \mathbb{R}^{n \times d}$; AE optimized projection matrix $\mathbf{W} \in \mathbb{R}^{d \times m}$
**Initialize:** Scalar vector $\boldsymbol{\lambda} \in \mathbb{R}^m$; Small positive float $\epsilon$
**Output:** Sorted AE projection matrix $\tilde{\mathbf{W}}$
 1: Normalize the feature matrix: $\mathbf{X_n} \leftarrow \mathbf{X}/\|\mathbf{X}\|$
 2: Compute data covariance matrix: $\boldsymbol{\Sigma} \leftarrow \mathbf{X_n}^T \mathbf{X_n}$
 3: Solve for $\boldsymbol{\lambda}$ such that $|\boldsymbol{\Sigma}\mathbf{W} - \boldsymbol{\lambda} \odot \mathbf{W}| \leqslant \epsilon$
 4: Sort column vectors $\boldsymbol{w} \in \mathbf{W}$ according to sorted indexation of $\boldsymbol{\lambda}$ to obtain $\tilde{\mathbf{W}}$

---

Practice guidelines and U.S. 21 CFR Part 50 (Protection of Human Subjects) and under the approval of Institutional Review Boards.

- **Parkinson's Progression Markers Initiative (PPMI)**: We pre-train the model on large-scale real-life Parkinsons Progression Markers Initiative (PPMI) data which is publicly available at [1] consists of 718 subjects, where 569 subjects are Parkinson's Disease (PD) patients and the rest 149 are Healthy Control (HC) ones. Eddy-current and head motion correction are performed using FSL[2] and the brain networks are extracted using the same tool. The EPI induced susceptibility artifacts correction is handled using Advanced Normalization Tools (ANT)[3]. In the meantime, 84 ROIs are parcelled from T1-weighted structural MRI using Freesurfer[4]. The brain networks are constructed using three whole brain tractography algorithms namely the Probabilistic Index of Connectivity (PICo), Hough voting (Hough), and FSL. Each resulted network for each subject is $84 \times 84$. Each brain network is normalized by the maximum value to avoid computation bias for the later feature extraction and evaluation, since matrices derived from different tractography algorithms differ in scales and ranges.
- **Bipolar Disorders (BP)**: This local and private dataset is composed of the resting-state fMRI and DTI image data of 52 Bipolar I subjects who are in euthymia and 45 Healthy Controls (HCs) with matched age and gender Cao et al. (2015); Ma et al. (2017). The fMRI data was acquired on a 3T Siemens Trio scanner using a T2* echo planar imaging (EPI) gradient-echo pulse sequence with integrated parallel acquisition technique (IPAT) and DTI data were acquired on a Siemens 3T Trio scanner. The brain networks are constructed using the CONN toolbox[5]. We performed the normalization and smoothing after first realigning and co-registering the raw EPI pictures. After that, the signal was regressed to remove the confounding effects of the motion artifact, white matter, and CSF. The 82 cortical and subcortical gray matter regions produced by Freesurfer were identified, and pairwise signal correlations were used to build the brain networks.
- **Human Immunodeficiency Virus Infection (HIV)**: Collected from the Early HIV Infection Study at Northwestern University, this private dataset involves fMRI and DTI brain networks for 70 subjects, with 35 of them early HIV patients and the other 35 Healthy Controls (HCs). These two groups of subjects do not differ in demographic distributions such as age and biological sex. The preprocessings for fMRI including brain extraction, slice timing correction and realignment are managed with the DPARSF[6] toolbox, while the preprocessings for DTI such as distortion correction are finished with the help of FSL[2] toolbox. Finally, brain networks with 90 regions of interest are constructed based on the automated anatomical labeling (AAL) Tzourio-Mazoyer et al. (2002).

## C  HYPERPARAMETER SETTINGS

**GNN Setup.** The GCN encoder is composed of 4 graph convolution layers with hidden dimensions of 32, 16, 16, and 8. Similarly, the GAT encoder is built from 4 graph attention layers with hidden dimensions of 32, 16, 16, and 8. Regarding GIN, which is slightly different, the encoder consists of 4 MLP layers with each MLP containing 2 linear layers with a unifying hidden dimension of 8.

---

[1] https://www.ppmi-info.org/
[2] https://fsl.fmrib.ox.ac.uk/fsl/fslwiki/
[3] http://stnava.github.io/ANTs/
[4] https://surfer.nmr.mgh.harvard.edu/
[5] http://www.nitrc.org/projects/conn/
[6] http://rfmri.org/DPARSF/

**Pre-training Pipeline Setup.** For two-level node contrastive sampling, we set $k = 2$ as the radius regarding $k$-hop neighborhood sampling for $\mathbf{S_1}$ and $\mathbf{S_4}$. To enable efficient computation on multi-graph MI evaluation, we resort to mini-batching and we set a default batch size of 32. In addition, we leverage the popular Adam Kingma & Ba (2015) optimizer with the learning rate set to 0.002 as well as the cosine annealing scheduler Loshchilov & Hutter (2017) to facilitate GNN training. In general, a complete pre-training cycle takes 400 epochs with an active deployment of early stopping.

**Atlas Mapping Regularizers Setup.** Following the discussion in section 3.3, the total running loss of the AE projection is given as:

$$\mathcal{L} = \mathcal{L}_{\text{rec}} + \alpha\mathcal{L}_{\text{loc}} + \beta\mathcal{L}_{\text{com}} + \gamma\mathcal{L}_{\text{KL}}, \tag{5}$$

in particular, we set $\alpha$, $\beta = 0.8$ and $\gamma = 0.01$. The one-layer AE encoder transforms the feature signals from all given datasets into a universally projected dimension of 32. Regarding the details of locality-preserving regularizer (*i.e.,* $\mathcal{L}_{\text{loc}}$), the transition matrix $\mathbf{T}$ is built from the 5-nearest-neighbor graph from the 3D coordinates of each atlas templates. For the sparsity-oriented regularizer (*i.e.,* $\mathcal{L}_{\text{KL}}$), the target sparsity value $\rho$ is set to $1e^{-5}$. The overall optimization process, which is similar to model pre-training, takes a total of 100 epochs with a learning rate of 0.02.

**Downstream Evaluation Setup.** For each target evaluation, the fine-tuning process features a 5-fold cross-validation, which approximately splits the dataset into 70% training, 10% validation, and 20% testing. To prevent model over-fitting, we implement a $L_2$ penalty with a coefficient of $1e^{-4}$. Overall, the model fine-tuning process, which is nearly identical to the other two training procedures, takes a total of 200 epochs with a learning rate of 0.001 and a cosine annealing scheduler.

## D ADDITIONAL EXPERIMENT

### D.1 DOWNSTREAM PERFORMANCE ON GAT AND GIN

**Table 4:** Disease prediction performance of our framework using GAT and GIN.

| Method | BP-fMRI | | BP-DTI | | HIV-fMRI | | HIV-DTI | |
|---|---|---|---|---|---|---|---|---|
| | ACC | AUC | ACC | AUC | ACC | AUC | ACC | AUC |
| Ours w/ GCN | **68.84**±8.26 | 68.45±8.96 | **66.57**±7.67 | **68.31**±9.39 | **77.80**±9.76 | 77.22±8.74 | **67.51**±8.67 | **67.74**±8.59 |
| Ours w/ GAT | 66.96±9.71 | 69.68±9.61 | 64.23±10.47 | 63.76±10.49 | 74.93±10.35 | 75.78±11.12 | 65.84±9.74 | 66.51±12.07 |
| Ours w/ GIN | 66.30±8.77 | **68.92**±9.37 | 64.48±9.83 | 66.44±8.58 | 75.96±9.56 | **77.63**±10.10 | 67.36±9.26 | 65.95±11.76 |

Table 4 reports the downstream performance of our full framework using GAT and GIN as backbone encoders. In general, the two encoders deliver inferior performance compared to GCN, which suggests that complex GNN convolutions (*e.g.,* GAT and GIN) might not be as effective as they seem when learning on bran network datasets.

### D.2 ADDITIONAL ABLATION STUDIES ON DTI VIEW

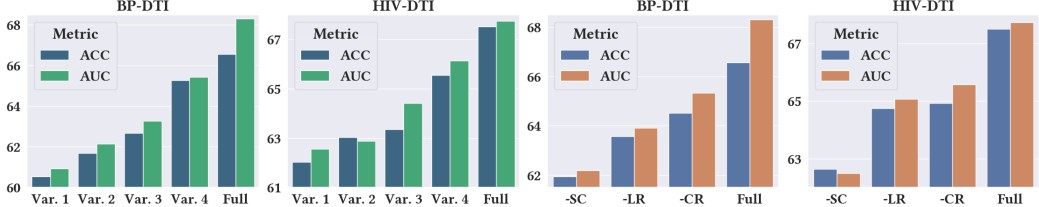

**Figure 7:** Additional ablation comparisons on DTI views. The left two subfigures refer to contrastive sampling considerations and the right two subfigures refer to atlas mapping regularizers. The $y$-axis refers to the numeric values of evaluated metrics (in %).

Figure 7 presents our ablation studies on the DTI view following the same setup as discussed in Section 4.2. We draw similar conclusions from the DTI-based analysis where each constituent component of our two-level sampling consideration as well as the atlas mapping mechanism has proven positive contribution and significance towards the overall performance and robustness.

