# OpenReview forum: "Pre-train Graph Neural Networks for Brain Network Analysis"
_ICLR.cc/2023/Conference — Submitted to ICLR 2023_

### Official Review · Reviewer_XzZd · 2022-10-20

**Confidence:** 5
**Correctness:** 3
**Technical Novelty And Significance:** 2
**Empirical Novelty And Significance:** 2
**Recommendation:** 3

**Clarity, Quality, Novelty And Reproducibility:**

- Quality of writing is quite poor and it is difficult to judge if this work is really novel or not. In a high-level sense, I think simply getting a good initialization doesn't sound very interesting but it can be better presented if mentioned as learning better initial representation.
- Reproducing the overall framework will not be easy based on the paper itself.


**Strength And Weaknesses:**

Strength:
- The proposed method can propose a globally sound initialization for a graph neural network (GNN) to be trained.
- The proposed method let one combine multiple datasets to come up with the initial representation.
- Experiments on various datasets demonstrate that the method improves training of other GNN module.

Weakness:
- The authors mentioned that obtaining a large-scale brain imaging dataset is difficult but the model is proposed assuming that a large-scale dataset exists for training the proposed method.
- Problem definition is not formally written. According to the introduction, as $D_i$ are all different, I assume that dataset sizes are different across $D_i$ but all of them are presumably given as $N$. It is not clear if there exists one $A_j$ per dataset, and the objective of GNN $f()$ (i.e., input and output) is also not clear. Moreover the description of $\theta^{t+1}$ seems not right with $L_{query_i} f(\theta\prime_i^t)$ as the $f()$ should have a graph $G$ as an input and multiplying L and f doesn't make sense; please correct me if I am wrong. Is there any description of measurement on the nodes?
- Lack of problem description lead to confusion in the method section. Contrastive learning is performed with $X$ but it is not clear what the $X$ is. Is this measurements on the graph nodes?
- At the beginning of 3.2, X was described as a sample. Then, $\bf S$ is named as an arbitrary sample but eq (2) sums over $\bf S$ where $X_S$ are its elements and not samples which is causing confusion.
- S1 and S4 are considered as positive samples but S4 is a subset of S3 which is a set of negative samples. Is there a reason why the data were divided into subsets such that they contradict to each other? Also, there is no equation or loss function explaining how these subsets are contrastively trained.
- It is not clearly explained how the authors deal with dataset heterogeneity.
- A summary table of individual dataset can be helpful.
- Ablation study is performed on the variants of subsets for contrastive learning. A reader perhaps would be more interested how much each regularization term designed by the author is contributing to the performance.


**Summary Of The Paper:**

The authors propose to leverage an unsupervised pre-training strategy to capture the intrinsic brain network structures regardless of specific clinical outcomes for obtaining GNN models that are easily adaptable to downstream tasks. The method utilizes large-scale brain imaging dataset that may not be related to a specific target task to improve the target task, and a data-driven parcellation atlas mapping pipeline is proposed to interpret knowledge transfer across studies with different ROI systems.

**Summary Of The Review:**

The proposed model has some merits as shown in the performance improvement but there are many places where the description of the framework are not very clear. The authors should better demonstrate their ideas such that this paper can be read and understood by a broader audience.

---

> ### Author Response · Authors · 2022-11-18
> **Authors' initial response to reviewer XzZd (2/2)**
>
> > Q5. It is not clearly explained how the authors deal with dataset heterogeneity.
>
> A: Thanks for the observation. Our work does not explicitly evaluate the heterogeneity of datasets. However, our proposed pre-training strategy is, in its nature, data agnostic since the goal is to learn a general parameter initialization from diverse sources of pre-training data.
>
>
> > Q6. An ablation study is performed on the variants of subsets for contrastive learning. A reader perhaps would be more interested in how much each regularization term designed by the author is contributing to the performance.
>
> A: We have actually compared how each regularization impacts the downstream performance in Section 4.2 on the right two sub-figures of Figure 4. Specifically, when analyzing the atlas regularizations, we fix the best contrastive sampling configuration. Similarly, when analyzing the effects of contrastive sampling variants, all regularization terms in the atlas mapping are used. Hence, we make sure that we only investigate the contribution of one of the design components in each ablation study.

---

> ### Author Response · Authors · 2022-11-18
> **Authors' initial response to reviewer XzZd (1/2)**
>
> > Q1. The authors mentioned that obtaining a large-scale brain imaging dataset is difficult but the model is proposed assuming that a large-scale dataset exists for training the proposed method.
>
> A: Thanks for the question. Obtaining large-scale datasets for local studies on a particular disease, such as BP and HIV in our experiments, is hard due to its difficulty in gathering enough (patient) samples and excessive cost in data pre-processing. Therefore, it is naturally hard to directly train a GNN classifier on these resource-limited local datasets. However, this does not mean that we are unable to find publicly available datasets at larger scales that are, in our case, used as sources to pre-train our GNN model.
>
>
> > Q2. The problem definition is not formally written.
>
> A: Thank you for the valuable feedback. We have made some modifications in the manuscript regarding the phrasings in the problem definition section, which are highlighted in red. Specifically, it is true that each dataset contains a varying number of samples. Thus, $N$ is now changed to $N_i$ to denote the unique sample size for each dataset $\mathcal{D}_i$. Similarly, the subscript describing each weighted graph object is now added with an additional index $i$ to highlight the dataset belonging of each graph. We also added an additional sentence describing the formal input and output of the GNN encoder $f_\theta(\cdot)$. In particular, the GNN output should be the learned node representations or a pooled graph-level representation. Regarding the joint loss function presented in the multi-dataset pre-training section (Sec. 3.1), we now change the equation to subscript the fast weights parameters. To clarify, the L and f are not in a multiplicative relationship. L simply means a calculated loss on the $i$-th query set using the encoder $f(\cdot)$ given its fast weight parameters $\theta^{t'}_i$.
>
>
> > Q3. The lack of problem description leads to confusion in the method section. Contrastive learning is performed with X but it is not clear what the X is. Are these measurements on the graph nodes?
>
> A: Thanks for the feedback. We have made certain changes in Equation 2 in Section 3.2 which are highlighted in red. Specifically, since using $X$ to denote node representations would lead to confusion with respect to the previous paragraph and Equation 1, a more accurate description should now be denoting these node representations in $z$ where $z_\alpha$ represents the anchor node’s representation, and the $\mathbf{S}$ now is a sampled set of other node representations used to evaluate their mutual information against the anchor. In later paragraphs, we then expand on how we define the different sampling strategies of $\mathbf{S_i}$ with respect to a given anchor, and we use the same formula in Equation 2 to evaluate mutual information scores.
>
> > Q4. S1 and S4 are considered positive samples but S4 is a subset of S3 which is a set of negative samples. Is there a reason why the data were divided into subsets such that they contradict each other? Also, there is no equation or loss function explaining how these subsets are contrastively trained.
>
> A: Thanks for the observation. We have made a slight change in our manuscript in Section 3.2 highlighted in red, where specifically in our setting, S1 and S4 are considered positive samples, while S2 and the set S3-S4 (i.e., S3 excluding S4) are negative samples. S4 is, indeed, a subset of S3, but it is only under our setting that we explicitly consider S4 as a unique sampling type. When S4 is not explicitly considered, which is the case in all other compared methods, S4 is inherently included in S3. When pre-training the neural network, the loss function is identical to the general contrastive learning objective described in Equation 1, where the goal is the minimize mutual information among negative pairs and maximize among positive pairs. Equation 2 explains how our method calculates the mutual information scores in detail. Therefore, the general loss function is nothing newer than combining Equation 1 and Equation 2.

---

### Official Review · Reviewer_khKm · 2022-10-25

**Confidence:** 4
**Correctness:** 3
**Technical Novelty And Significance:** 3
**Empirical Novelty And Significance:** 3
**Recommendation:** 5

**Clarity, Quality, Novelty And Reproducibility:**

#### Clarity
- Does the number of subjects in the dataset reflects the number of images? And it is not clear how cross-validation has been performed. It must be performed subject-wise to ensure that the subject is only in one subset.

#### Quality
- There are some issues with technical sophistication as there is no hold-out set and no methodology to evaluate the data-driven atlas mapping.
- The authors should address huge variance issues in the performance metrics.

#### Novelty
- The proposed solution seems novel.

#### Reproducibility
- The statistical significance testing details are missing.


**Strength And Weaknesses:**

#### Strength
- The experiments were performed on multiple datasets.
- There are multiple graph baselines.
- There are ablation studies for choosing contrastive sampling strategies and atlas mapping regularizes.

#### Weaknesses
- It is not clear what statistical significance test has been used. The t-test should not be used as you can use it only if you pass the normality assumption. Further, you need to report statistics in a valid format. In most scenarios in Table 2 and Table 4, the variance is so high that most performance metrics should overlap, so I don't think the metrics should pass the significance testing. In addition, have you explored the reason behind such a huge variance?
- There is no final holdout-test set, only 5-fold cross-validation. The validation set has to be used only to select checkpoints or hyperparameters.
- Alignment of the virtual ROIs is only done visually. Consider segmentation measures (e.g., DICE (F1)) or other measurable methodology to evaluate the performance of data-driven brain atlas mapping concerning the original atlas.
- There are no non-graph-based baselines or discussions.
- There is no discussion on related work for Neuroimaging domain:
  - That describes the use of GNNs or pretraining:
    - Mahmood, Usman, et al. "Multi network InfoMax: A pre-training method involving graph convolutional networks." arXiv preprint arXiv:2111.01276 (2021)
    - Mahmood, Usman, et al. "Deep Dynamic Effective Connectivity Estimation from Multivariate Time Series." arXiv preprint arXiv:2202.02393 (2022).
  - That describes data-driven brain atlas mapping (e.g., via spectral clustering):
    - Geenjaar, Eloy, et al. "Spatio-temporally separable non-linear latent factor learning: an application to somatomotor cortex fMRI data." arXiv preprint arXiv:2205.13640 (2022)
- Could you clarify how the matching of the nodes for a positive pair between the source datasets is enforced? It seems that nodes should be paired based on spatial alignment. How do you guarantee that nodes are spatially aligned to the same brain region after the autoencoder?
- I am unsure if you used the same MAML framework for all the baselines. Could it be the main reason for the improvement? Could you compare your model by pretraining on two datasets and perform transfer learning by fine-tuning on the third?







**Summary Of The Paper:**

The manuscript proposed data-driven brain atlas mapping and contrastive pretraining for brain network analysis. The authors show improved results over previous baselines, ablate the proposed method based on different strategies, and perform a visual examination of ROI alignment. Contrastive learning is based on Jensen-Shannon divergence with positive and negative pairs defined based on the graph. The data-driven brain atlas mapping incorporates an autoencoder with brain-oriented regularizers based on locality, brain modules, and sparsity. In addition, variance-based dimension sorting is proposed to align ROIs of the different dataset and their atlases.

**Summary Of The Review:**

Overall, I like the proposed approach and comparison with multiple baselines. The proposed framework seems to improve over previous baselines. However, I have some minor issues that I want to clarify.

Further, I think the authors should propose some methodology for the data-driven atlas mapping, as the visual evaluation does not seem enough.

---

> ### Author Response · Authors · 2022-11-18
> **Authors' initial response to reviewer khKm (2/2)**
>
> > Q3. Alignment of the virtual ROIs is only done visually. Consider segmentation measures (e.g., DICE (F1)) or other measurable methodology to evaluate the performance of data-driven brain atlas mapping concerning the original atlas.
>
> A: Thanks for the constructive and valuable suggestion, we will experiment with more evaluation metrics regarding atlas mapping, such as DICE (F1) as mentioned above, in our future elaborations.
>
> > Q4. There are no non-graph-based baselines or discussions.
>
> A: Thanks for the suggestion. Our work focuses on pre-training graph neural networks for brain network learning, so we do not think it is necessary for us to compare non-graph-based models. Moreover, many of the graph-based models we compared have been established based on comparison with non-graph-based models in their original papers already.
>
> > Q5. There is no discussion on related work for Neuroimaging domain: (1) That describes the use of GNNs or pretraining (2) That describes data-driven brain atlas mapping (e.g., via spectral clustering)
>
> A: Thanks for the suggestions. Regarding GNN application in neuroimaging domain, we have cited the following sources [1-3] in the introduction, methodology, and experiment sections. To the best of our knowledge, we are among the first to formulate a multi-task pre-training objective for GNN learning on brain networks. However, the work mentioned here [4] operates in a pertinent setting but experiments on a vastly different design and data type. Regarding data-driven atlas mapping, since our setting does not restrict nor consider the input modality, the framework is motivated such that it can be generically extended to all fully-processed brain network objects. In addition, we are the first to formulate our atlas transformation into a multi-dataset feature alignment objective.
>
> [1] Cui, H., et al. Braingb: A benchmark for brain network analysis with graph neural networks. TMI 2022.
> [2] Li X., et al. Braingnn: Interpretable brain graph neural network for fMRI analysis. Med Image Anal, 2021b.
> [3] Martensson, G., et al.Stability of graph theoretical measures in structural brain networks in Alzheimer's disease. Scientific reports, 8:1–15, 2018.
> [4] Usman, M,. et al. "Multi-network InfoMax: A pre-training method involving graph convolutional networks." arXiv preprint arXiv:2111.01276 (2021).
>
> > Q6. Could you clarify how the matching of the nodes for a positive pair between the source datasets is enforced? It seems that nodes should be paired based on spatial alignment. How do you guarantee that nodes are spatially aligned to the same brain region after the autoencoder?
>
> A: We do not sample positive or negative pairs across datasets. The two-level node sampling only samples positive and negative pairs across multiple graphs within a dataset. During the pre-training phase, although multiple datasets are fed into the framework, the objectives are computed on each dataset separately. Therefore, spatial alignment or misalignment of nodes across datasets makes no impact on the entire pre-training pipeline. Furthermore, since data objects within a dataset use a fixed node system, the node orderings, and spatial matching are naturally aligned, and such alignment is not influenced by an autoencoder (which is basically a fixed feature projection function applied to each graph).
>
>
> > Q7. I am unsure if you used the same MAML framework for all the baselines. Could it be the main reason for the improvement? Could you compare your model by pretraining on two datasets and perform transfer learning by fine-tuning on the third?
>
> A: Thanks for the question. Yes, the MAML multi-task pre-training technique is used for all methods reported in Table 2 Section 4.1 except the non-pre-train (NPT) baseline. As can be seen from the table, the MAML technique does not necessarily make improvements, especially on non-GNN-based methods. Hence, the improvement margin shown in the empirical studies mainly focuses on contributions from the novel two-level contrastive node sampling strategy. The goal of our model pre-training is to learn a general parameter initialization instead of transferring the knowledge in the source dataset since there is no working evidence on data similarity between our source and target data and more importantly, the unsupervised pre-training does not contain relevant tasks knowledge that is applicable to a supervised downstream objective. In addition, effective pre-training, like in many other contexts, requires sufficient training data. In our experimental setting, only the PPMI dataset (which is publicly accessible) is suitable to be used as a pre-train source.
>
>
> > Q8. Does the number of subjects in the dataset reflect the number of images?
>
> A: Yes. The number of subjects in each dataset represents the number of available brain networks. This number can be smaller than the number of raw images due to removal for quality control.

---

> > ### Comment · Reviewer_khKm · 2022-11-19
> > **Response to Q3-Q8**
> >
> > Q4: I suggest considering such works. It will only strengthen your work, as it shows your methods with respect to different approaches.
> >
> > Q5: Spectral clustering will be highly relevant for data-driven atlas mapping as it can be used for images and time series. It is a universal approach. In contrast, your method is tied to some form of input parcellation, which does not make it universal for a new dataset if it has a different parcellation. In addition, the dataset should still be preprocessed for your case. I also think that parcellation might be of a different kind based on anatomy or function.
> >
> > Q7:
> > - If the MAML technique "does not necessarily make improvements," do we actually need multi-task then? It would be a great idea to show that different parcellations can help to learn better representation.
> > - If the goal of "pre-training is to learn a general parameter initialization," then you need to show that your model is useful and can be generalized. You will need to consider some target parcellation and see whether your network transfers using (e.g., linear evaluation, partial fine-tuning, low-shot fine-tuning, full fine-tuning).

---

> > > ### Author Response · Authors · 2022-11-21
> > > **Thanks for the further suggestions**
> > >
> > > Dear reviewer,
> > >
> > > Q4: Our goal in this work is not to establish better brain network models, or promote graph-based models over non-graph-based ones. Instead, we are just providing a better pre-training framework for graph-based models. Thus we believe the most important point of our experiments is to show that our proposed pre-training framework can indeed improve the graph-based models. We can consider using more graph-based-models to show this in the future, but we still don't think there is a need to compare with non-graph-based ones, and comparing with them would be trivial anyway as the advantages of graph-based models over non-graph-based ones have been established in many existing works already.
> > >
> > > Q7: We thank the reviewer for the suggestion and would consider adding more ablation studies to support the effectiveness of MAML or remove it at all in the future. We will also consider more specific generalization evaluations as suggested.
> > >
> > > Thanks again for all the helpful suggestions.

---

> ### Author Response · Authors · 2022-11-18
> **Authors' initial response to reviewer khKm (1/2)**
>
> > Q1. It is not clear what statistical significance test has been used. The t-test should not be used as you can use it only if you pass the normality assumption. Further, you need to report statistics in a valid format. In most scenarios in Table 2 and Table 4, the variance is so high that most performance metrics should overlap, so I don't think the metrics should pass the significance testing. In addition, have you explored the reason behind such a huge variance?
>
> A: Firstly, assuming normality for experimental results and applying t-tests is only too commonly used in existing research, and we almost never see anyone do a  “normality check” for this.  Secondly, high variance just does not mean not passing the t-test– p-values are decided by the sample size, mean difference, and sample variance, not only sample variance. In fact, we used the paired t-test to conduct our significance testing with a significance level of alpha = 0.05. When the results from our method are compared to specific baselines, all pairs have passed the significance testing with the maximum t-statistics reported to be 0.042.  Finally, the issue of high variance is a common phenomenon in brain network predictions  [1,2]. Compared with the reported results in some of the existing works, our variances are actually smaller possibly because of the effect of pre-training.
>
> [1] Cui, H., Dai, W., Zhu, Y., Li, X., He, L., & Yang, C. (2022). Interpretable graph neural networks for connectome-based brain disorder analysis. In International Conference on Medical Image Computing and Computer-Assisted Intervention (pp. 375-385). Springer, Cham.
> [2] Kong, Z., Sun, L., Peng, H., Zhan, L., Chen, Y., & He, L. (2021). Multiplex graph networks for multimodal brain network analysis. arXiv preprint arXiv:2108.00158.
>
>
> > Q2. There is no final holdout-test set, only 5-fold cross-validation. The validation set has to be used only to select checkpoints or hyperparameters.
>
> A: Thanks for pointing out this aspect. The hyperparameter setting for GNN learning in our work is adapted from standard designs introduced in related efforts such as [1-5]. Hence, we do not explicitly select hyperparameters based on an exhaustive grid search on a validation set. The 5-fold cross-validation, in our setting, is used to directly evaluate test performance on 5 different stratified splits.
>
> [1] Chang, Qi, et al. "Classification of first-episode schizophrenia, chronic schizophrenia and healthy control based on brain network of mismatch negativity by graph neural network." IEEE Transactions on Neural Systems and Rehabilitation Engineering (2021).
> [2] Wein, Simon, et al. "A graph neural network framework for causal inference in brain networks." Scientific reports (2021).
> [3] Yang, Chunde, et al. "Autism spectrum disorder diagnosis using graph attention network based on spatial-constrained sparse functional brain networks." Computers in Biology and Medicine (2021).
> [4] Hu, Jinlong, et al. "GAT-LI: a graph attention network based learning and interpreting method for functional brain network classification." BMC Bioinformatics (2021).
> [5] Li, Xiaoxiao, et al. "Braingnn: Interpretable brain graph neural network for fMRI analysis." Medical Image Analysis (2021).

---

> > ### Comment · Reviewer_khKm · 2022-11-19
> > **Q1 and Q2**
> >
> > Q1: I'm afraid I have to disagree with this idea. The Wilcoxon signed-rank test is often used because it does not require normality assumptions. Further, I suggest checking the ICML 2022 tutorial on this if you want to see better practices. https://www.cl.uni-heidelberg.de/statnlpgroup/empirical_methods_tutorial/
> >
> > Q2: I'm afraid I also have to disagree with this. In the current setup, technical sophistication would be lacking. You should have some form of hold-out set that you haven't used for anything. Otherwise, It is possible to cherry-pick the checkpoints that perform well on validation. Further, I suggest having ablations to ensure you found an excellent set of hyperparameters for your method.

---

> > > ### Author Response · Authors · 2022-11-19
> > > **Thanks for the further suggestions**
> > >
> > > Dear reviewer,
> > >
> > > Q1: Thanks for pointing out the Wilcoxon test. Our point here is, t-test is also commonly used for analyzing the significance of experimental results, and we don't think using it instead of Wilcoxon test means the improvements we observed in the experiments are all invalid. We will consider the better practices you pointed out next time.
> > >
> > > Q2: Our whole dataset is split for 5 times, each time with 4/5 as training and 1/5 as test. We abused the term of "cross-validation" and called these test sets "validation sets". This is a misuse of the terminology. However, we have made it clear in the answer above that we have not used this "validation" set for anything other than testing. We will follow your suggestions to create another validation set for more careful hyperparameter tuning next time.
> > >
> > > Thank you again for all of the helpful suggestions.

---

> ### Comment · Reviewer_khKm · 2022-11-19
> **Rebuttal decision**
>
> I want to thank the authors for the rebuttal response. I decided to keep my score. Authors should improve technical sophistication and sharpen their contributions with decisive discussion.

---

> > ### Author Response · Authors · 2022-11-21
> > **Thanks for reading and acknowledging our rebuttal**
> >
> > Dear reviewer,
> >
> > We want to thank you for reading our rebuttal and providing more detailed suggestions! Although we cannot fully agree with all of your arguments, we can understand most of your concerns and the justifiable overall rating of a borderline rejection. Also, we really appreciate many of your helpful suggestions and will certainly leverage them to further improve our work substantially.

---

### Official Review · Reviewer_TtZF · 2022-10-25

**Confidence:** 5
**Correctness:** 1
**Technical Novelty And Significance:** 1
**Empirical Novelty And Significance:** Not applicable
**Recommendation:** 3

**Clarity, Quality, Novelty And Reproducibility:**

The authors seem to lack some knowledge about brain disease and neuroimaging data analysis, so many disease-related analyses are inaccurate. For example, in the second sentence of the abstract, the authors state that ” Recent studies in neuroscience and neuroimaging analysis have reached a consensus that interactions among brain regions of interest (ROIs) are driving factors for neural development and disorders.” It is not true. There is still much debate about the role of neuroimaging, especially functional brain networks based on fmri in brain disease. Conclusions from existing MRI imaging works are also difficult to replicate.(Botvinik-Nezer, R., Holzmeister, F., Camerer, C.F. et al. Variability in the analysis of a single neuroimaging dataset by many teams. Nature 582, 84–88 (2020). https://doi.org/10.1038/s41586-020-2314-9)
In 3.3.2, the authors state that “Therefore, it is reasonable to assume a shared virtual ROI system underlying different parcellation systems”. There is no basis for this statement. The authors believe that different diseases and different parcellation systems may share the same virtual ROI system. This hypothesis is difficult to be convinced unless the author has a large amount of clinical data to support it.


**Strength And Weaknesses:**

The second work of this paper about brain atlas mapping is problematic. The author states that they need to convert networks between different ROI systems because the brain networks in the two datasets will have different numbers and semantics of nodes. However, to train the model, you do not need to convert the network between different brain templates, you only need to generate data with the same brain template. Because raw MRI image data is voxel-based, an easy way is to generate a brain network based on the same brain template. So, I do not see a reasonable motivation for using deep learning methods to solve these simple problems. The authors may argue that the available data are based on different brain templates and need to convert. However, keep in mind that brain networks are just an analytical representation of neuroimaging data, and different networks can be generated based on different templates and parameters. Therefore, it is simpler and more accurate to generate a unified brain network directly from raw image data using the same brain template.

**Summary Of The Paper:**

This paper presents an unsupervised GNN pre-training technique for brain networks. The authors first designed a two-level contrastive learning method and then propose a data-driven atlas mapping technique for mapping different ROI systems. The method was evaluated in three brain network datasets. The results showed the proposed method outperformed other pretraining frameworks in disease classification problems.

**Summary Of The Review:**

The second motivation of this paper is problematic, so it is not suitable for published in ICLR.

---

> ### Author Response · Authors · 2022-11-18
> **Authors' initial response to reviewer TtZF (2/2)**
>
> > Q3. In 3.3.2, the authors state that “Therefore, it is reasonable to assume a shared virtual ROI system underlying different parcellation systems”. There is no basis for this statement. The authors believe that different diseases and different parcellation systems may share the same virtual ROI system. This hypothesis is difficult to be convinced unless the author has a large amount of clinical data to support it.
>
> A: Thanks for pointing this out. We argue about the assumption of a shared virtual ROI system from the perspective of anatomical functional brain modules [1-3] that are extensively studied in brains. These functional modules provide a higher level of organization of the brain surface. On the other hand, the different ROI parcellations are more granular and specific to different analytical purposes. Our assumption, in its nature, does not concern itself with a particular disease or clinical data, but a general hypothesis that, regardless of the parcellation system used, there exists a mapping from it to the bigger functional modules.
>
> [1] Philipson, Lars. “Functional modules of the brain”. Journal of Theoretical Biology, 2002
> [2] Anderson JR, Bothell D, Byrne MD, Douglass S, Lebiere C, Qin Y. “An integrated theory of the mind.” Psychological Review. 2004
> [3] Hilger, K., Fukushima, M., Sporns, O., & Fiebach, C. J. “Temporal stability of functional brain modules associated with human intelligence.” Human brain mapping. 2020

---

> ### Author Response · Authors · 2022-11-18
> **Authors' initial response to reviewer TtZF (1/2)**
>
> > Q1. However, to train the model, you do not need to convert the network between different brain templates, you only need to generate data with the same brain template. Because raw MRI image data is voxel-based, an easy way is to generate a brain network based on the same brain template. So, I do not see a reasonable motivation for using deep learning methods to solve these simple problems. The authors may argue that the available data are based on different brain templates and need to convert. However, keep in mind that brain networks are just an analytical representation of neuroimaging data, and different networks can be generated based on different templates and parameters. Therefore, it is simpler and more accurate to generate a unified brain network directly from raw image data using the same brain template.
>
> A: This work focuses on the pre-training of GNN models for brain network analysis, and our goal is to effectively leverage existing large-scale brain network datasets by designing appropriate pre-training techniques. In such a setting, the existing brain network datasets are given as they are, and pre-processing raw neural imaging data using one fixed atlas is not an option. In fact, the pre-processing of raw imaging data into brain networks is an extremely hard and time-consuming process, which requires extensive domain expertise and manual inspections at multiple steps, and that is out of the scope of this paper. Meanwhile, requiring everyone to create their brain network datasets using the same fixed atlas is also impossible.
>
> > Q2. In the second sentence of the abstract, the authors state that ” Recent studies in neuroscience and neuroimaging analysis have reached a consensus that interactions among brain regions of interest (ROIs) are driving factors for neural development and disorders.” It is not true. There is still much debate about the role of neuroimaging, especially functional brain networks based on fMRI in brain disease. Conclusions from existing MRI imaging works are also difficult to replicate. (Botvinik-Nezer, R., Holzmeister, F., Camerer, C.F. et al. Variability in the analysis of a single neuroimaging dataset by many teams. Nature 582, 84–88 (2020). https://doi.org/10.1038/s41586-020-2314-9)
>
> A: Brain connection analysis is still an open and active research domain in neuroscience. In recent years, the analysis of brain networks has become increasingly popular in neuroimaging studies in order to understand human brain organization across different groups of individuals [1,2,3,4,5,6]. There have been abundant findings in neuroscience research showing that neural circuits largely cause and define brain functions and disease [7,8,9,10]. In this paper, we focus on developing a machine learning algorithm to facilitate knowledge discovery from brain network data. Whether brain network data itself is the absolutely right tool for all clinical analysis is really out of our concern. We are confused if the reviewer is implying that all analytical tools on brain network are useless until one can prove brain network to be the best way to model brains?
>
> [1] Su, C., Xu, Z., Pathak, J., & Wang, F. (2020). Deep learning in mental health outcome research: a scoping review. Translational Psychiatry, 10(1), 1-26.
> [2] Satterthwaite, T. D., Wolf, D. H., Roalf, D. R., Ruparel, K., Erus, G., Vandekar, S., ... & Gur, R. C. (2015). Linked sex differences in cognition and functional connectivity in youth. Cerebral cortex, 25(9), 2383-2394.
> [3] Wang, Y., Kang, J., Kemmer, P. B., & Guo, Y. (2016). An efficient and reliable statistical method for estimating functional connectivity in large scale brain networks using partial correlation. Frontiers in neuroscience, 10, 123.
> [4] Deco, G., Jirsa, V. K., & McIntosh, A. R. (2011). Emerging concepts for the dynamical organization of resting-state activity in the brain. Nature Reviews Neuroscience, 12(1), 43-56.
> [5] Wang, S., He, L., Cao, B., Lu, C. T., Yu, P. S., & Ragin, A. B. (2017, August). Structural deep brain network mining. In Proceedings of the 23rd ACM SIGKDD international conference on knowledge discovery and data mining (pp. 475-484).
> [6] Yu, R., Qiao, L., Chen, M., Lee, S. W., Fei, X., & Shen, D. (2019). Weighted graph regularized sparse brain network construction for MCI identification. Pattern recognition, 90, 220-231.
> [7] Insel, T. R., & Cuthbert, B. N. (2015). Brain disorders? precisely. Science, 348(6234), 499-500.
> [8] Williams, L. M. (2016). Precision psychiatry: a neural circuit taxonomy for depression and anxiety. The Lancet Psychiatry, 3(5), 472-480.
> [9] Li, W., Wang, M., Li, Y., Huang, Y., & Chen, X. (2016). A novel brain network construction method for exploring age-related functional reorganization. Computational intelligence and neuroscience, 2016.
> [10] Lynn, C. W., & Bassett, D. S. (2019). The physics of brain network structure, function and control. Nature Reviews Physics, 1(5), 318-332.

---

### Official Review · Reviewer_Y7QB · 2022-11-01

**Confidence:** 3
**Correctness:** 3
**Technical Novelty And Significance:** 2
**Empirical Novelty And Significance:** 2
**Recommendation:** 5

**Clarity, Quality, Novelty And Reproducibility:**

Quality:
It is clear the authors have put a lot into the evaluation of the proposed method. Some implementation and dataset details are in the appendix. However, there are no citations for the datasets used in the main text (and limited info is found in appendix)- are these public or private, where are they from, were they properly collected with IRB approval, etc. At a minimum citation for the datasets need to be included in the main paper if they have already been previously introduced elsewhere.

Clarity:
The organization of sections is good, and the writing is generally ok. However there are some details that need to be clarified (see above weaknesses).
Some additional notes on writing: the references are all within the text which is difficult to read - they should be in parenthetical form (eg \citep)

Originality:
The contribution for the GNN pretraining is that the contrastive learning framework considers both node and graph level information, using nodes in a k-hop neighborhood of the target node i in other graphs in the dataset as positive samples. Other parts of the pretraining are from prior work (MAML optimization, definitions of pos/neg sample within graph, cost function).
I think the originality of the transformation mapping portion lies in the specific combination of regularizers applied, as I believe these have been used before.


**Strength And Weaknesses:**

Strengths:
1. The method considers leveraging additional brain network information not previously considered in GNN pretraining - using samples from corresponding node positions in other graphs as positive examples for the anchor node. This seems like a reasonable way to define more positive samples utilizing other graphs for contrastive learning.

2. The approach considers the problem of combining brain imaging datasets that used different parcellations and thus have different graph node configurations.

3. The experiments have meaningful comparisons to baselines separated into different groups (e.g., no pretraining, baselines using different levels of node/graph representation in contrastive learning) to help understand the improvements in model decision.

4. There are several ablation experiments included to see the effect of different parts of the design approach.


Weaknesses:
1. The motivation for the data driven atlas transformation is to get a common graph configuration for all the datasets, since different datasets may use different parcellations. However, what if one fixed atlas were used and applied to the existing imaging data - this is not compared? Is the idea that it is not always available? Furthermore, my understanding is each dataset transformation is learned separately - then how can the transformed space be aligned? I understand that the "virtual ROIs" are set according to some variance ranking, but this does not mean the ROIs would be aligned across datasets - as seen in Fig. 6 which is supposed to demonstrate overlap, I would argue that the ROIs do NOT look aligned. Then, if these ROIs are not aligned, does the pretraining of a model to use on a different dataset make sense?

2. It is unclear to me if the atlas transformation is learned separately from the pretraining step? And whether each transformation mapping is learned separately for each dataset? Also, is it based on all the data, or trained using just training data? It would be unfair using all data as the representation is then fit to the test data as well.

3. In comparing the proposed method to other methods with pretraining, I presume this is with the proposed atlas transformation, while the comparison method does not have any such mapping estimated - is this correct? Or are other methods also learned using the atlas transformation? If not, then it is unclear the contribution of the proposed contrastive learning formulation vs. the atlas transformation. If could perform the atlas transformation for comparison methods, then can more fairly compare just the contrastive learning for pretraining. Or, do not include any atlas transformation, and then compare the contrastive learning approaches.

4. Some of the experimental details are confusing to me. For example, the plots in Fig. 5 show pretraining and learning on the task dataset for 150 epochs, while the appendix says pretraining goes for 400 epochs and the task learning for 200 epochs. In Fig. 5b, it seems that the curves for all the approaches may not have converged yet, so the comparison is not quite fair, even though at that epoch (which is not 200) the proposed method is most accurate.

5. At a higher level, I wonder whether expecting the same parcellation of brain regions for fMRI vs DTI data makes sense - it may be that the natural groupings based on fMRI data may not align with natural groupings based on structural DTI data. Yet both these types of datasets are simply combined for analysis. Could the authors comment on this?



**Summary Of The Paper:**

This paper proposes a method for graph-based learning from brain imaging datasets, where they 1) pretrain GNN using 2-level contrastive learning, which leverages some brain network knowledge, under a MAML optimization approach , and 2) perform a data-driven atlas transformation with various regularizations and variance sorting to provide some alignment of ROIs across different datasets. The method is tested using 3 datasets, where the large PPMI dataset is used for the pretraining and the transfer to task learning is tested on the other 2 datasets. Experiments are performed comparing multiple baselines with and without pretraining and studying effect of model components through ablation studies.

**Summary Of The Review:**

While the work largely builds off of existing prior work, the new approach to defining positive samples for contrastive learning of GNNs for brain datasets is interesting and makes sense, and a number of experiments have been performed to demonstrate the potential advantage of the proposed methods. However, some clarifications are needed.

---

> ### Author Response · Authors · 2022-11-18
> **Authors' initial response to reviewer Y7Q8 (2/2)**
>
> > Q4. Some of the experimental details are confusing to me. For example, the plots in Fig. 5 show pretraining and learning on the task dataset for 150 epochs, while the appendix says pretraining goes for 400 epochs and task learning for 200 epochs. In Fig. 5b, it seems that the curves for all the approaches may not have converged yet, so the comparison is not quite fair, even though at that epoch (which is not 200) the proposed method is most accurate.
>
> A: Thanks for the questions and also the detailed observation. The pre-training process takes, indeed, a maximum of 400 epochs. However, in the appendix, we also mentioned active employment of early stopping, meaning that we stop the training loop when the reported loss fluctuates in a stalemate without showing a significant decrease. This situation generally occurs between the first 100 to 200 epochs. Hence, it would be unnecessary to keep training until meeting the maximum 400-epoch limit. However, it is a good suggestion to further expand the training epochs a bit in Figure 5 because the current ones do not show clear convergence of all compared models. Therefore, we have updated the subfigures (a) and (b) in Figure 5 to include 200 epochs instead of 150 in the revision.
>
>
> > Q5. At a higher level, I wonder whether expecting the same parcellation of brain regions for fMRI vs DTI data makes sense - it may be that the natural groupings based on fMRI data may not align with natural groupings based on structural DTI data. Yet both these types of datasets are simply combined for analysis. Could the authors comment on this?
>
> A: Thanks for bringing up this question. Actually how to choose an ideal brain parcellation for different modalities is still an open question that is actively studied by the community [1, 2]. Though different methods have been developed for constructing brain parcellations using different imaging modalities, for the several mainstream brain parcellations, there is no consensus on whether a specific atlas is more favorable to one modality than the other. It is also common in previous literature to parcellate neuroimaging from different modalities using the same atlas [3, 4]. Besides, there have also been efforts in standardizing human brain parcellation across modalities based on transformations [5].
>
> [1] Eickhoff, S. B., Yeo, B. T., & Genon, S. (2018). Imaging-based parcellations of the human brain. Nature Reviews Neuroscience, 19(11), 672-686.
> [2] Moghimi, P., Dang, A. T., Netoff, T. I., Lim, K. O., & Atluri, G. (2021). A Review on MR Based Human Brain Parcellation Methods. arXiv preprint arXiv:2107.03475.
> [3] McKenna, B. S., Theilmann, R. J., Sutherland, A. N., & Eyler, L. T. (2015). Fusing fMRI and DTI Measures of Brain Function and Structure to Predict Working Memory and Processing Speed Performance among Inter-episode Bipolar Patients. Journal of the International Neuropsychological Society: JINS, 21(5), 330.
> [4] Ma, G., He, L., Lu, C. T., Shao, W., Yu, P. S., Leow, A. D., & Ragin, A. B. (2017, November). Multi-view clustering with graph embedding for connectome analysis. In Proceedings of the 2017 ACM on Conference on Information and Knowledge Management (pp. 127-136).
> [5] Lawrence, R. M., Bridgeford, E. W., Myers, P. E., Arvapalli, G. C., Ramachandran, S. C., Pisner, D. A., ... & Vogelstein, J. T. (2021). Standardizing human brain parcellations. Scientific
>
>
> > Q6. However, there are no citations for the datasets used in the main text (and limited info is found in the appendix)- are these public or private, where are they from, were they properly collected with IRB approval, etc?
>
> A:  Thanks for the valuable suggestions. PPMI is a public dataset available at https://www.ppmi-info.org/, while HIV is collected from the Early HIV Infection Study at Northwestern University. Both HIV and BP are private datasets. The data processed is anonymous with no personally identifiable information. All studies are conducted according to the Good Clinical Practice guidelines and U.S. 21 CFR Part 50 (Protection of Human Subjects) and under the approval of Institutional Review Boards. They have also been studied in previous literature [1, 2]. We have further detailed the dataset information in the revised manuscript highlighted in red.
>
> [1] Liu, Y., He, L., Cao, B., Yu, P., Ragin, A., & Leow, A. (2018, April). Multi-view multi-graph embedding for brain network clustering analysis. In Proceedings of the AAAI Conference on Artificial Intelligence (Vol. 32, No. 1).
> [2] Zhang, X., He, L., Chen, K., Luo, Y., Zhou, J., & Wang, F. (2018). Multi-view graph convolutional network and its applications on neuroimage analysis for Parkinson's disease. In AMIA Annual Symposium Proceedings (Vol. 2018, p. 1147). American Medical Informatics Association.

---

> ### Author Response · Authors · 2022-11-18
> **Authors' initial response to reviewer Y7Q8 (1/2)**
>
> > Q1. The motivation for the data-driven atlas transformation is to get a common graph configuration for all the datasets, since different datasets may use different parcellations. However, what if one fixed atlas were used and applied to the existing imaging data - this is not compared? Is the idea that it is not always available? Furthermore, my understanding is each dataset transformation is learned separately - then how can the transformed space be aligned? I understand that the "virtual ROIs" is set according to some variance ranking, but this does not mean the ROIs would be aligned across datasets - as seen in Fig. 6 which is supposed to demonstrate overlap, I would argue that the ROIs do NOT look aligned. Then, if these ROIs are not aligned, does the pretraining of a model to use on a different dataset make sense?
>
> A: Thanks for the questions.
>
> First, this work focuses on the pre-training of GNN models for brain network analysis, and our goal is to effectively leverage existing large-scale brain network datasets by designing appropriate pre-training techniques. In such a setting, the existing brain network datasets are given as they are, and pre-processing raw neural imaging data using one fixed atlas is not an option. In fact, the pre-processing of raw imaging data into brain networks is an extremely hard and time-consuming process, which requires extensive domain expertise and manual inspections at multiple steps, and that is out of the scope of this paper. Meanwhile, requiring everyone to create their brain network datasets using the same fixed atlas is also impossible.
>
> Second, the method we designed to align the atlases without supervision is based on the assumption that they can be aligned according to the sample variances. One can always argue that the alignment does not look perfect through visual inspection unless they look exactly the same. However, there are indeed certain similarities regarding the highlighted regions across datasets, and we just did not perform exhaustive cherry-picking in order to show the most “perfect” alignments. Even though, our quantitative experimental results in Figure 4 do support the effectiveness of our proposed variance-based atlas mapping techniques.
>
>
> > Q2. It is unclear to me if the atlas transformation is learned separately from the pretraining step? And whether each transformation mapping is learned separately for each dataset? Also, is it based on all the data, or trained using just training data? It would be unfair to use all data as the representation is then fit to the test data as well.
>
> A: Thanks for the question. Yes, atlas transformation is applied before GNN pre-training and is learned separately for each dataset. It is done only using the training data– the autoencoder models are trained only on the training data, and then applied to the test data.
>
>
> > Q3. In comparing the proposed method to other methods with pretraining, I presume this is with the proposed atlas transformation, while the comparison method does not have any such mapping estimated - is this correct? Or are other methods also learned using the atlas transformation? If not, then it is unclear the contribution of the proposed contrastive learning formulation vs. the atlas transformation. If could perform the atlas transformation for comparison methods, then can more fairly compare just the contrastive learning for pretraining. Or, do not include any atlas transformation, and then compare the contrastive learning approaches.
>
> A: Thanks for the question. For results in Table 2, the same full version of atlas mapping is applied for all compared models, because none of them would work without atlas mapping across the datasets (for example, because the node features are of different dimensions). This also justifies our motivation of doing atlas mapping in the first place. Moreover, in our ablation studies (Figure 4), we have tried to clearly separate the benefit of atlas mapping and contrastive learning. When we compare different contrastive learning variants, we fixed the same full version of atlas mapping. When we compare different atlas mapping variants, we fixed the same full version of contrastive learning.

---

### Decision · Program_Chairs · 2023-01-20

**Decision:**

Reject

**Justification For Why Not Higher Score:**

All reviewers recommend rejection.

**Justification For Why Not Lower Score:**

N/A

**Metareview: Summary, Strengths And Weaknesses:**

Summary:
This paper suggests a method for pre-training GNNs for brain connectivity using a) Using 2-level contrastive learning to pretrain GNNs using prior knowledge about brain networks, and b) performing a data-driven atlas transformation to provide alignment across datasets.


Strengths:
The use of the contrastive learning to encode prior knowledge seems novel and useful


Weaknesses:
The second alignment part of the algorithm seems unnecessarily complicated and weakly motivated -- why did the authors not just use a single atlas for alignment? There were seeveral concerns regarding clarity. Finally, the neuroscience hypotheses and conclusions drawn were found to be problematic in comparison with the existing literature.